# Beyond Proxy Metrics: A New Evaluation Framework for LLM Compression by Directly Measuring Generative Faithfulness

## Abstract

Current evaluation methods for Large Language Model (LLM) compression, which rely on proxy metrics like perplexity and curated benchmarks, often correlate poorly with real-world generative performance. This discrepancy creates a significant gap between reported scores and practical utility. To address this, we introduce a new evaluation framework that dispenses with such proxies by directly measuring a compressed model's generative faithfulness to its uncompressed counterpart on real-world user queries. The core of our framework is Conditional Generation Accuracy (CGA), a novel metric that employs a teacher-forcing paradigm to assess the compressed model's ability to replicate the original model's next-token prediction at each step, conditioned on the ground-truth prefix. This approach effectively avoids the cascading errors that confound traditional text-similarity measures. We apply this framework to a comprehensive evaluation of nine mainstream compression methods across models from 7B to 32B parameters and context lengths up to 24K tokens. Our results establish a clear performance hierarchy and reveal distinct scaling laws with respect to model size and context length. For instance, while most methods' performance improves with model size, we find that quantization and KV cache dropping degrade with longer contexts, whereas a sparse attention baseline uniquely improves. Our work provides a more rigorous and reliable foundation for benchmarking LLM compression. To promote transparent and reproducible progress, we have open-sourced our benchmark code and will launch a leaderboard.

## 1 Introduction

The rapid adoption of Large Language Models (LLMs) has created an urgent demand for efficient inference under constrained GPU resources (Li et al., 2024a; Qu et al., 2025; Zheng et al., 2025). Model compression has emerged as a promising solution (Zhu et al., 2024; Park et al., 2024; Xu et al., 2024a), with many training-free or calibration-light approaches (Frantar et al., 2022; Lin et al., 2024; Frantar & Alistarh, 2023) claiming substantial inference speedups with only marginal accuracy loss. However, a significant gap exists between these claims and real-world performance. Our qualitative analysis (Appendix B) reveals that compressed models, such as Qwen2.5-7B-Instruct (Yang et al., 2025) with AWQ (Lin et al., 2024), are prone to degradation, including repetition, nonsensical outputs, and impaired instruction following capabilities. Crucially, these failures are consistently missed by standard benchmarks (Wang et al., 2024; Xu et al., 2024b; Wang et al., 2025), which often report the performance of these methods as nearly lossless. This stark discrepancy highlights the urgent need to re-evaluate the current evaluation methods for LLM compression.

Current evaluation methods predominantly rely on Perplexity (PPL) and Question Answering (QA) benchmarks (Jaiswal et al., 2024; Wang et al., 2024; Clark et al., 2019), both of which suffer from fundamental limitations.

**Limitations of Perplexity.** Perplexity suffers from two key limitations. First, its unbounded value range and model-specific baselines make fair cross-model comparison difficult (Xu et al., 2024c). More importantly, Perplexity can be easily exploited. Certain compression strategies, such as KV cache dropping (Li et al., 2024b; Zhang et al., 2023), can be over-optimized to achieve favorable Perplexity scores while producing outputs that diverge significantly from the original model, thereby undermining practical usability.

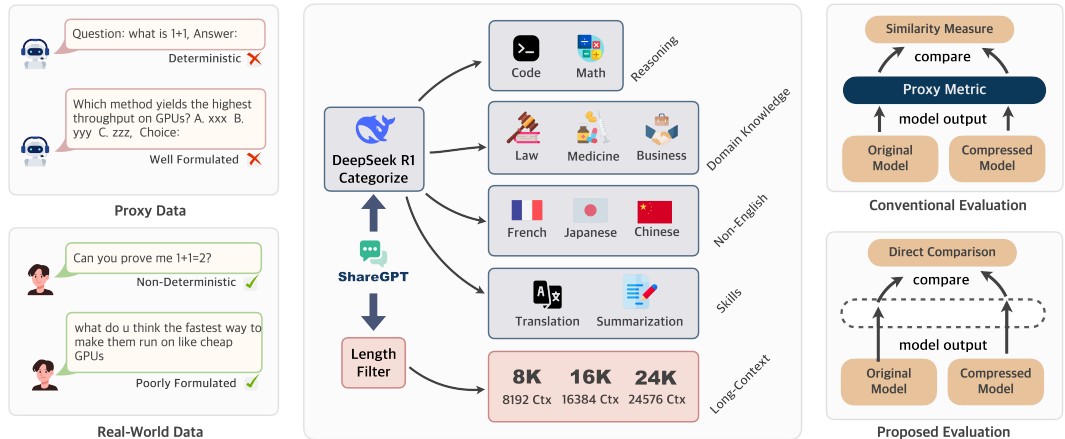

Figure 1: *(Left)* We replace curated benchmarks with open-ended, real-world user queries. *(Center)* These queries are sourced from ShareGPT and categorized for fine-grained analysis. *(Right)* We measure generative faithfulness by directly comparing the outputs of the compressed model to the original, replacing conventional proxy metrics.

**Limitations of QA Benchmarks.** The shift to downstream QA benchmarks (Jaiswal et al., 2024; Hong et al., 2024) introduces its own set of issues. These benchmarks typically rely on rigid, closed-form formats (*e.g.*, multiple-choice questions) that are fundamentally misaligned with the open-ended generative tasks common in real-world usage. This focus on final answer accuracy, rather than the generative process, creates a significant distributional gap between benchmark data and authentic model chats. Consequently, models risk overfitting to benchmark schemas, leading to evaluation scores that do not reliably predict real-world performance.

The primary objective of compression should be to preserve the original model's generative behavior, a principle we define as generative faithfulness or fidelity. However, existing evaluation paradigms do not measure this fidelity directly. They instead incentivize the optimization of indirect scores, creating a critical disconnect where methods are tuned for benchmark performance rather than for replicating the original model. This fosters a misleading research landscape where reported gains fail to translate into practical utility, thereby slowing down meaningful progress.

To address this issue, we introduce a novel evaluation framework (Figure 1) that eliminates proxies in favor of direct, fidelity-based evaluation using real-world user prompts.

To measure fidelity, we propose Conditional Generation Accuracy (CGA). CGA sidesteps the cascading errors of edit distance (Bar-Yossef et al., 2004) and the external model bias of BERTScore (Zhang et al., 2019) by employing a teacher-forcing paradigm. Specifically, we evaluate the compressed model's next token prediction accuracy at each step, conditioned on the golden context generated by the original model. This provides a direct and unbiased assessment of the compressed model's ability to replicate the original model's generative behavior.

Next, we replace synthetic benchmarks (Jaiswal et al., 2024; Hong et al., 2024) with real-world user queries sourced from ShareGPT. This dataset reflects the open-ended and often ambiguous nature of real-world interactions, where no single "correct" answer exists. Our framework handles such data naturally by treating the original model's output as the reference. For more granular analysis, we leverage DeepSeek R1 (Guo et al., 2025) to classify queries by domain (*e.g.*, math, code) and stratify them by context length.

We applied our framework to conduct a comprehensive evaluation of 9 mainstream compression methods. Our experiments span multiple model scales, assessing Qwen2.5-Instruct (Yang et al., 2025) from 7B to 32B parameters and context lengths up to 24K tokens. Our framework generates normalized fidelity scores within [0, 1] that are directly comparable, leading to several key insights.

To promote transparency and accelerate progress in the field, we will release a public leaderboard upon paper acceptance. We hope this work provides a rigorous, interpretable, and scalable foundation for fair benchmarking in LLM compression research.

**Key Takeaways**

**Comparative Performance**

- Clear Hierarchy: A distinct performance hierarchy emerges among compression categories: Low-Precision Attention > INT4 Quantization > 50% Pruning > KV Cache Dropping.

- Quantization: Within INT4 methods, GPTQ (Frantar et al., 2022) consistently shows higher fidelity and better generalization than AWQ (Lin et al., 2024).

- Sparse Attention: A sparse attention baseline (Top-10%) unexpectedly outperforms all evaluated INT4 quantization methods.

- KV Cache Dropping: These methods (Li et al., 2024b; Zhang et al., 2023) perform very poorly. Their outputs are often misaligned with the original model, raising serious questions about their practical viability.

**Scaling with Model Size**

- Favorable Scaling: The fidelity of quantization (Frantar et al., 2022; Lin et al., 2024) and most low-precision attention (Shah et al., 2024; Zhang et al., 2024) methods improves significantly on larger models.

- Consistent Scaling: Pruning methods (Frantar & Alistarh, 2023; Sun et al., 2023b) and Top-10% sparse attention exhibit stable performance across different model sizes.

- Negative Scaling: KV cache dropping methods (Li et al., 2024b; Zhang et al., 2023) show uniquely degraded performance as model size increases.

**Scaling with Long Contexts**

- Favorable Scaling: Top-10% sparse attention is the only tested method whose fidelity robustly improves with longer contexts.

- Consistent Scaling: Pruning based methods (Frantar & Alistarh, 2023; Sun et al., 2023b) and FlashAttention FP8 (Shah et al., 2024) maintain consistent performance across all tested context lengths.

- Negative Scaling: All INT4 quantization (Frantar et al., 2022; Lin et al., 2024) and KV cache dropping methods (Li et al., 2024b; Zhang et al., 2023) show a marked decline in fidelity as context length grows.

## 2 RELATED WORKS

**LLM Compression.** Deploying large language models (LLMs) on resource-constrained hardware has spurred extensive research into model compression (Zhu et al., 2024; Park et al., 2024; Xu et al., 2024a). Key lossy techniques include low-precision attention (Shen et al., 2021; Han et al., 2023), quantization (Lang et al., 2024; Liu et al., 2024), pruning (Ma et al., 2023; Sun et al., 2023a), and KV cache dropping (Liu et al., 2023; Kang et al., 2024; Hooper et al., 2024; Dong et al., 2024). State-of-the-art methods in each category often report substantial efficiency gains with minimal performance degradation.

*Low-Precision Attention.* Specialized hardware, such as NVIDIA's Hopper architecture, has enabled FP8 precision for accelerating attention mechanisms (Choquette, 2022). The FP8 implementation of FlashAttention (Dao et al., 2022; Dao, 2023; Shah et al., 2024), an IO-aware algorithm, yields significant throughput gains but can introduce noise that degrades performance on sensitive downstream tasks (Chen et al., 2024). To mitigate this, hybrid-precision methods like SageAttention (Zhang et al., 2024) selectively use integer formats (*e.g.*, INT8 or INT4) for query and key (QK) projections while retaining FP8 for value and output (VO) computations to preserve accuracy. Another approach involves sparsity. Methods like MoBA (Lu et al., 2025) and NSA (Yuan et al., 2025) use heuristics to approximate attention by processing a subset of KV blocks. Although practical, their performance is generally upper-bounded by Top-K Attention (Gupta et al., 2021), which serves as a benchmark by selecting the most relevant KV blocks based on true attention scores.

*Quantization.* Quantization is a fundamental technique for LLM compression (Lang et al., 2024; Liu et al., 2024; Egiazarian et al., 2024). GPTQ (Frantar et al., 2022), a prominent post-training

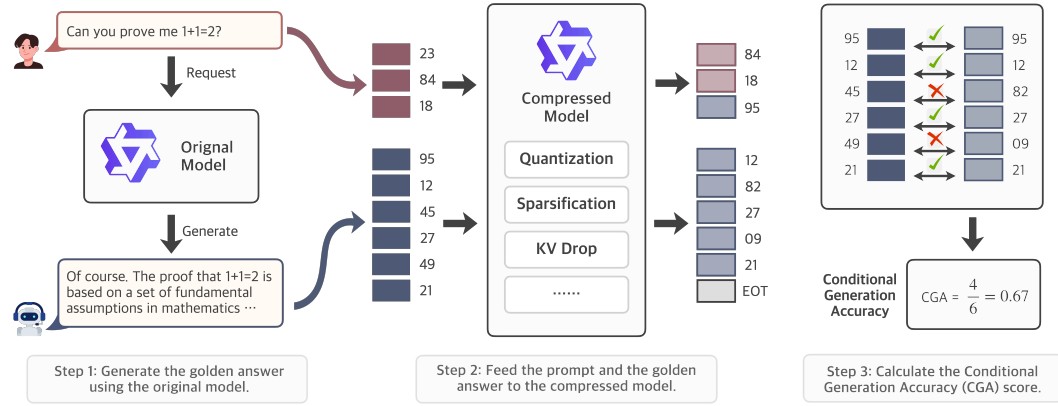

Figure 2: An illustration of the Conditional Generation Accuracy (CGA) calculation process. First, the original model generates a reference output sequence for a given prompt. Next, the compressed model predicts each token conditioned on the reference sequence from the original model. Finally, the CGA score is computed as the fraction of tokens correctly predicted by the compressed model.

quantization (PTQ) method, uses approximate second-order Hessian information for layer-wise weight quantization, achieving high accuracy at low bit-widths like INT4 and INT3. However, its quantization process is computationally expensive. In contrast, AWQ (Lin et al., 2024) offers a faster alternative by scaling weights based on activation magnitudes prior to quantization. This reliance on activation statistics may render AWQ more sensitive to the calibration data distribution, potentially impacting its generalization (Hubara et al., 2021).

*Pruning.* Pruning improves efficiency by removing redundant model parameters (Ma et al., 2023; Sun et al., 2023a). While simple magnitude-based pruning is a common baseline, it risks removing weights critical to model performance (Lee et al., 2020; Sun et al., 2023a). More sophisticated methods address this limitation. SparseGPT (Frantar & Alistarh, 2023) employs a calibration process to attain high sparsity with minimal accuracy loss. Wanda (Sun et al., 2023a) further refines this by considering both weight magnitude and the norms of corresponding input activations, providing a more robust metric for a weight's contribution.

*KV Cache Dropping.* Several methods compress the KV cache to accelerate generative inference (Liu et al., 2023; Kang et al., 2024; Hooper et al., 2024; Dong et al., 2024). H2O (Zhang et al., 2023) identifies and retains influential "heavy-hitter" KV pairs based on cumulative attention scores while evicting less important ones to adhere to a fixed budget. Extending this, SnapKV (Li et al., 2024b) also compresses the prompt's KV cache to further reduce memory and computational overhead.

**Compressed LLM Evaluation.** The evaluation methodology for compressed LLMs has evolved rapidly. Early work relied on perplexity (PPL) (Xu et al., 2024c), a metric now widely criticized for its poor correlation with downstream task performance and its susceptibility to being over-optimized (Fang et al., 2024).

This led to a shift towards comprehensive question-answering (QA) benchmarks (Xu et al., 2024b; Wang et al., 2025). Researchers have proposed complex evaluation suites assessing dimensions like fairness and ethics (Hong et al., 2024; Xu et al., 2024b), alongside large-scale leaderboards aggregating numerous QA tasks (Jaiswal et al., 2024). However, a primary limitation is their typical reliance on structured formats like multiple-choice questions (Myrzakhan et al., 2024). This format poorly reflects open-ended, real-world user interactions, introducing evaluation bias and creating the risk of over-optimizing compression methods for specific test formats (Kumar et al., 2025).

Motivated by the limitations of proxy metrics, a concurrent work directly compares the outputs of compressed and original models (Wang et al., 2025). This approach has explored metrics such as edit distance (Bar-Yossef et al., 2004) and BERTScore (Zhang et al., 2019). Both have notable drawbacks. Edit distance is unreliable for long generations, as minor initial errors can cascade into large, misleading divergence scores. BERTScore, while designed for semantic similarity, introduces potential biases from its own underlying pretrained model (Sun et al., 2022).

Table 1: Validation of evaluation metrics on Qwen2.5-7B-Instruct (Yang et al., 2025). While proxy metrics like PPL and MMLU (Hendrycks et al., 2021) often fail to reflect true performance degradation (misleading scores are underlined), our proposed CGA score aligns closely with the human-aligned DeepSeek Score, providing a more faithful measure of generative quality.

| Metric | INT4 Quant | | Low-Precision Attn | | | KV Cache Drop | | Pruning | | Baseline |
| | AWQ | GPTQ | Flash FP8 | Sage | Top-10% | SnapKV | H2O | Sparse | Wanda | |
|---|---|---|---|---|---|---|---|---|---|---|
| WT2 PPL | 17.76 | 16.79 | 116.27 | 16.00 | 15.99 | 15.57 | 18.56 | 21.37 | 20.89 | 15.99 |
| MMLU | 0.718 | 0.718 | 0.570 | 0.718 | 0.718 | 0.401 | 0.680 | 0.651 | 0.658 | 0.719 |
| DeepSeek Score | 0.833 | 0.849 | 0.042 | 0.882 | 0.865 | 0.031 | 0.000 | 0.289 | 0.303 | 1.0 |
| WT2 PPL (norm) | 0.172 | 0.454 | **0.000** | 0.997 | 1.000 | **1.000** | 0.077 | 0.005 | 0.008 | 1.0 |
| MMLU (norm) | **0.999** | **0.999** | 0.793 | **0.999** | **0.999** | 0.558 | 0.946 | 0.907 | 0.915 | 1.0 |
| CGA (ours) | 0.916 | 0.922 | 0.586 | 0.987 | 0.953 | 0.538 | 0.465 | 0.784 | 0.737 | 1.0 |

## 3 FIDELITY-BASED EVALUATION FRAMEWORK

The central goal of evaluating a compressed LLM is to quantify its faithfulness to the original model. Let $\mathcal{F}$ denote an original, uncompressed LLM and $\tilde{\mathcal{F}}$ represent its compressed counterpart. For any given user prompt $X$, these models generate output sequences $Y = \mathcal{F}(X)$ and $\tilde{Y} = \tilde{\mathcal{F}}(X)$, respectively. An ideal evaluation framework would assess the quality of $\tilde{\mathcal{F}}$ by directly comparing $\tilde{Y}$ to $Y$ across a distribution of prompts $X$ that accurately reflects real-world usage.

Instead of directly comparing the generated output sequences $Y$ and $\tilde{Y}$, prevailing methods (Xu et al., 2024b; Hong et al., 2024; Jaiswal et al., 2024; Wang et al., 2025) resort to proxy metrics (*e.g.*, PPL and QA benchmarks). The final score $S$ in these frameworks is not a direct measure of generative fidelity but an indirect comparison of abstract values derived from the model outputs. For PPL, the score compares perplexity values rather than the generated text itself:

$$S_{PPL} \propto \sum_{X_{PPL}} \text{sim}\left(\text{PPL}(\mathcal{F}(X_{PPL})), \text{PPL}(\tilde{\mathcal{F}}(X_{PPL}))\right) \quad (1)$$

Likewise, QA benchmarks (Xu et al., 2024b; Hong et al., 2024; Jaiswal et al., 2024) compare final task scores, abstracting away the generative process entirely and focusing only on the outcomes:

$$S_{QA} \propto \sum_{X_{QA}} \mathbf{1}\left(\text{score}(\mathcal{F}(X_{QA})) = \text{score}(\tilde{\mathcal{F}}(X_{QA}))\right) \quad (2)$$

The reliance on these proxies fundamentally obscures the evaluation. A compressed model may achieve a PPL or QA score nearly identical to the original's while generating text that is factually incorrect, stylistically divergent, or simply incoherent. This indirection is the primary reason for the well-documented gap between benchmark performance and real-world user experience.

To remedy these issues, we propose a new evaluation framework based on two core principles: using real-world data distribution and a direct fidelity-based metric.

### 3.1 ALIGNING WITH REAL-WORLD DATA DISTRIBUTIONS

To ground our evaluation in practical applications, we constructed a dataset from real-world user interactions sourced from ShareGPT, a public repository of diverse and multilingual user prompts. For granular analysis, we structured this dataset in two ways.

**Domain-Specific Subsets.** We employed DeepSeek R1 (Guo et al., 2025) to automatically classify prompts into ten distinct categories: two reasoning tasks (math, code), three domain-knowledge areas (business, law, medicine), three non-English languages (Japanese, French, Chinese), and two specific skills (summarization, translation). Each category contains approximately 100 challenging prompts, enabling a targeted assessment of model performance in specialized areas. Examples for each subtask are provided in Appendix C.

**Long-Context Subsets.** To rigorously analyze performance in long-context scenarios, a critical use case for compressed models, we segmented the data into three length-based tiers: 8K, 16K, and 24K tokens. Each tier contains approximately 20 prompts, forming a dedicated testbed to simulate the demands of long-context applications.

This structured approach ensures our evaluation is aligned with genuine user queries while enabling precise, fine-grained analysis of model performance across various domains and context lengths.

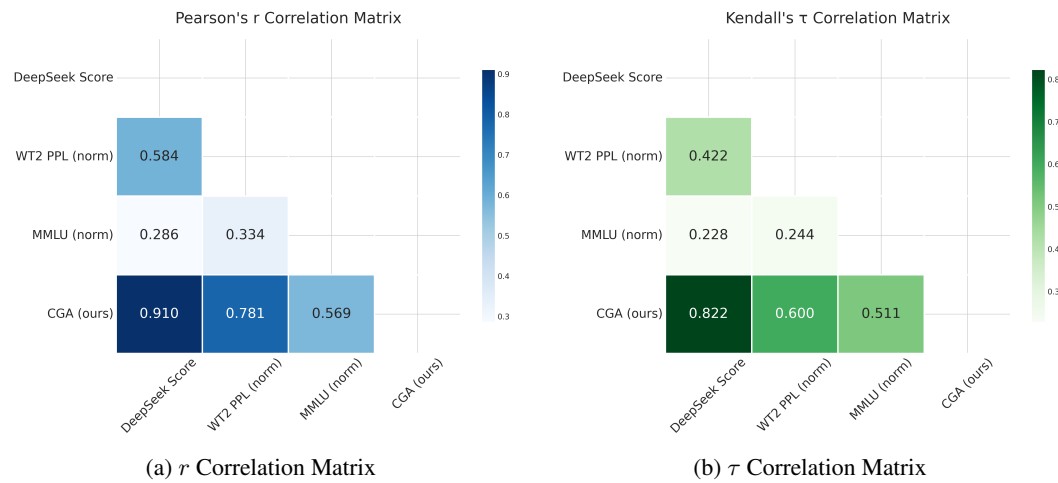

(a) $r$ Correlation Matrix $\qquad$ (b) $\tau$ Correlation Matrix

Figure 3: Correlation matrices of evaluation metrics. (a) $r$ coefficients, measuring linear relationships. (b) $\tau$ coefficients, assessing monotonic relationships. Darker cells indicate stronger positive correlation. Our proposed CGA consistently demonstrates the highest correlation ($r = 0.910$, $\tau = 0.822$) with human preference (DeepSeek Score).

## 3.2 CONDITIONAL GENERATION ACCURACY

The ultimate measure of a compressed model's quality is its ability to faithfully replicate the generative behavior of the original model. Existing metrics fail to capture the nuanced aspects of generation, including tone, style, and the step-by-step reasoning process embedded in an output sequence. To overcome this, we introduce Conditional Generation Accuracy (CGA), a metric designed to directly compare the output distributions of the compressed and original models at the token level.

The standard auto-regressive generation process for the original model $\mathcal{F}$ and its compressed model $\tilde{\mathcal{F}}$ at a given step $i$ is defined as:

$$y_i = \mathcal{F}(X, y_1, \ldots, y_{i-1}) \quad \text{and} \quad \tilde{y}_i = \tilde{\mathcal{F}}(X, \tilde{y}_1, \ldots, \tilde{y}_{i-1}), \tag{3}$$

A naive comparison between the final sequences $Y$ and $\tilde{Y}$ (*e.g.*, using edit distance) is unreliable. A single-token divergence early in the generation process can cause subsequent outputs to diverge completely, leading to a cascade of errors that incorrectly penalizes the compressed model for what may be a minor initial deviation. This accumulation of errors renders simple sequence-level similarity metrics ineffective for evaluating long-form generation.

To isolate the predictive capability of the compressed model at each step, our CGA metric adopts a teacher-forcing paradigm. Instead of conditioning the compressed model's next-token prediction on its own, potentially flawed, previously generated tokens ($\tilde{y}_{<i}$), we condition it on the ground-truth prefix generated by the original model ($y_{<i}$):

$$\hat{y}_i = \tilde{\mathcal{F}}(X, y_1, y_2, \ldots, y_{i-1}). \tag{4}$$

Here, $\hat{y}_i$ represents the compressed model's most likely next token given the ideal context from the original model. By decoupling the evaluation at each step from errors made in prior steps, this approach eliminates the issue of cascading errors and provides a direct measure of the compressed model's ability to replicate the original model's output at every point in the generation process.

The final CGA score is the mean accuracy over all tokens in the generated sequence, averaged across all prompts in the evaluation dataset $D$:

$$S_{\text{CGA}} = \frac{1}{|D|} \sum_{X \in D} \left( \frac{1}{|Y|} \sum_{i=1}^{|Y|} \mathbf{1}(\hat{y}_i = y_i) \right) = \frac{1}{|D|} \sum_{X \in D} \left( \frac{1}{|Y|} \sum_{i=1}^{|Y|} \mathbf{1}\left( \tilde{\mathcal{F}}(X, y_{<i}) = \mathcal{F}(X, y_{<i}) \right) \right). \tag{5}$$

As the formulation shows, CGA directly measures the alignment between the compressed and original models' next-token predictions, conditioned on the golden output history. It uses no external models or abstract proxies, providing the most direct and fundamental assessment of generative faithfulness.

Table 2: Performance comparison of nine compression methods on Qwen2.5-7B-Instruct, measured by CGA. Cell backgrounds are colored on a gradient, with red indicating lower fidelity scores.

| Category | Sub-dataset | Compression Method | | | | | | | | |
|---|---|---|---|---|---|---|---|---|---|---|
| | | AWQ | GPTQ | Flash FP8 | Sage | Top-10% | SnapKV | H2O | Sparse | Wanda |
| Reasoning | code | 0.9228 | 0.9504 | 0.4576 | 0.9844 | 0.9472 | 0.7466 | 0.6323 | 0.8150 | 0.8165 |
| | math | 0.9567 | 0.9426 | 0.5457 | 0.9829 | 0.9671 | 0.6350 | 0.6193 | 0.6455 | 0.8752 |
| Knowledge | fact | 0.8773 | 0.8789 | 0.6078 | 0.9874 | 0.9010 | 0.6403 | 0.5217 | 0.7855 | 0.7486 |
| | law | 0.9023 | 0.9055 | 0.1899 | 0.9850 | 0.9205 | 0.6593 | 0.5782 | 0.7635 | 0.7755 |
| | business | 0.8994 | 0.9102 | 0.6753 | 0.9861 | 0.9401 | 0.6456 | 0.5929 | 0.7511 | 0.7700 |
| | medicine | 0.9169 | 0.9299 | 0.4307 | 0.9795 | 0.9619 | 0.7007 | 0.6070 | 0.7929 | 0.8063 |
| Non-English | french | 0.9165 | 0.9319 | 0.3199 | 0.9874 | 0.9425 | 0.7171 | 0.5794 | 0.7706 | 0.7811 |
| | chinese | 0.8986 | 0.9094 | 0.6236 | 0.9866 | 0.9291 | 0.6876 | 0.4982 | 0.7289 | 0.7496 |
| | japanese | 0.8914 | 0.8935 | 0.7486 | 0.9874 | 0.9254 | 0.6608 | 0.5301 | 0.8547 | 0.8827 |
| Skills | en2zh | 0.9497 | 0.9341 | 0.7273 | 0.9874 | 0.9801 | 0.4152 | 0.4812 | 0.8086 | 0.8313 |
| | zh2en | 0.9383 | 0.9608 | 0.7158 | 0.9809 | 0.9824 | 0.4800 | 0.4647 | 0.8415 | 0.8939 |
| | sum | 0.8491 | 0.9176 | 0.8710 | 0.9837 | 0.9243 | 0.1948 | 0.7156 | 0.8026 | 0.7999 |
| **Average** | | 0.9091 | 0.9212 | 0.5796 | 0.9860 | 0.9428 | 0.5997 | 0.5709 | 0.7797 | 0.8200 |

# 4 EXPERIMENTS

We proceed in four stages. First, we validate that our Conditional Generation Accuracy (CGA) metric aligns more closely with human-aligned judgments than widely used proxy metrics (Section 4.1). Second, we conduct a comprehensive comparison of all selected compression methods to establish a clear performance hierarchy (Section 4.2). Finally, we investigate how the fidelity of each method scales with two critical factors: model size, from 7B to 32B parameters (Section 4.3), and context length, from 8K to 24K tokens (Section 4.4).

## 4.1 VALIDATING THE CGA

The foundational step of our analysis is to validate that CGA serves as a more reliable indicator of generative faithfulness than conventional proxy metrics. To do this, we benchmarked CGA against Perplexity, QA benchmarks, and DeepSeek judgments, using the latter as a high-quality approximation for human preference.

**Evaluation Setup.** We evaluate our proposed evaluation metric, *CGA*, against three distinct types of metrics: Perplexity, QA Score, and DeepSeek Score.

- *Perplexity (PPL).* We measure the average Perplexity on the WikiText-2 dataset (Merity et al., 2017). To ensure a fair comparison, the raw PPL scores are normalized to a [0, 1] range. This is achieved by applying a sigmoid function to the difference in PPL between the compressed and the baseline models.

- *QA Score.* We use the MMLU benchmark (Hendrycks et al., 2021) as a representative question answering task. The evaluation is conducted in a zero-shot setting using the `lm-eval-harness`. The final score is reported as the ratio of the compressed model's accuracy to the original model's accuracy, normalizing the result to a [0, 1] interval.

- *DeepSeek Score.* To approximate human evaluation, we employ DeepSeek R1 (Guo et al., 2025) for a pairwise comparison of outputs generated by the original and compressed models in response to a given prompt. To mitigate positional bias, the presentation order of the model outputs is randomized. The final score is calculated as the win rate of the compressed model against the original model.

**Results.** As shown in Table 1, conventional proxy metrics demonstrate significant limitations. The normalized Perplexity score shows poor alignment with the human-aligned DeepSeek Score, incorrectly assigning a high score to SnapKV (Li et al., 2024b), a method exhibiting severe performance collapse, while penalizing effective methods like FlashAttention FP8 (Shah et al., 2024). The MMLU score, while able to identify the poorest performing methods, lacks discriminatory power for more effective techniques. Its scores saturate near 1.0, failing to distinguish between methods with different fidelity levels, such as AWQ (Lin et al., 2024) and SageAttention (Zhang et al., 2024).

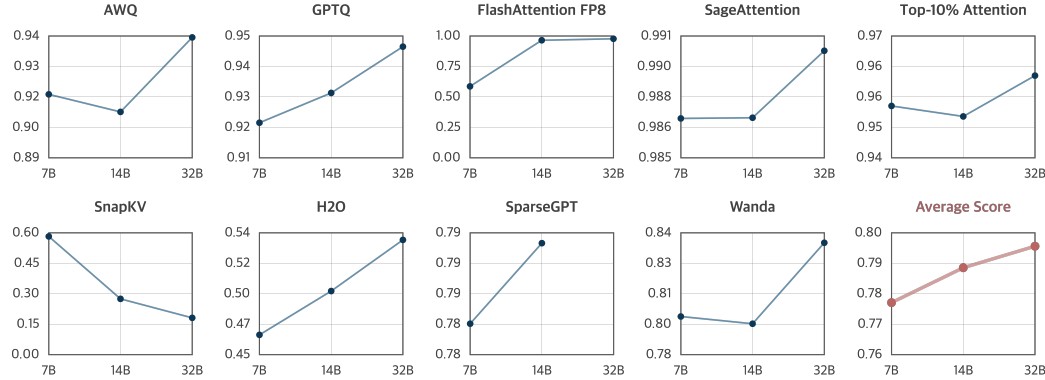

Figure 4: Scaling of compression method fidelity with model size. Most methods improve on larger models, indicating that greater parameter redundancy enhances compressibility.

The correlation analysis in Figure 3 provides quantitative support for these observations. CGA achieves a strong Pearson correlation ($r = 0.910$) and Kendall rank correlation ($\tau = 0.822$) with the DeepSeek Score, indicating a significantly better alignment with human-aligned judgments compared to PPL and MMLU. While the DeepSeek Score is a valuable validation tool, its inherent variance and high inference cost make it impractical for scalable benchmarking. CGA, in contrast, offers a robust, deterministic, and highly correlated alternative.

## 4.2 COMPARATIVE ANALYSIS OF COMPRESSION METHODS

We conducted a large-scale benchmark of nine compression methods on Qwen2.5-7B-Instruct (Yang et al., 2025) to establish a clear performance hierarchy. The detailed results, presented in Table 2, provide a granular comparison across various domains and tasks. Analyses for the 14B and 32B models are included in Appendix D.

Our analysis reveals a distinct performance hierarchy among the compression categories. Low-precision attention methods achieve the highest fidelity, followed by INT4 quantization and 50% pruning. Within the top-performing category, the hybrid-precision approach of SageAttention (Zhang et al., 2024) proves highly effective, mitigating the significant performance degradation seen in a naive FP8 implementation like FlashAttention (Shah et al., 2024). Notably, the Top-10% sparse attention baseline delivered surprisingly strong performance, ranking second overall despite a 90% effective sparsity and outperforming all evaluated INT4 quantization methods. This result highlights the significant potential of structured sparsity as a high-fidelity compression strategy. Conversely, KV cache dropping methods (Li et al., 2024b; Zhang et al., 2023) performed very poorly, with CGA scores at a default 20% budget low enough to cast serious doubt on their practical viability.

## 4.3 SCALING WITH MODEL SIZE

A central hypothesis in model compression is that larger models, with their inherent parameter redundancy, should be more resilient to information loss. To test this, we evaluated how the fidelity of each compression method scales with model size, applying them to the 7B, 14B, and 32B versions of the Qwen2.5-Instruct model (Yang et al., 2025).

The results, plotted in Figure 4, largely confirm this hypothesis. Most methods, including quantization (Frantar et al., 2022; Lin et al., 2024) and low-precision attention (Shah et al., 2024; Zhang et al., 2024), show improved fidelity on larger models. This trend empirically supports the idea that the impact of a fixed compression ratio diminishes as model capacity grows. While some methods show a minor dip at the 14B scale, the overall trend remains positive. SnapKV (Li et al., 2024b) is a notable exception, exhibiting negative scaling where fidelity degrades as the model grows larger.

## 4.4 SCALING WITH CONTEXT LENGTH

Many compression techniques, particularly low-precision attention (Shah et al., 2024; Zhang et al., 2024) and KV cache dropping (Li et al., 2024b; Zhang et al., 2023), are motivated by the need to

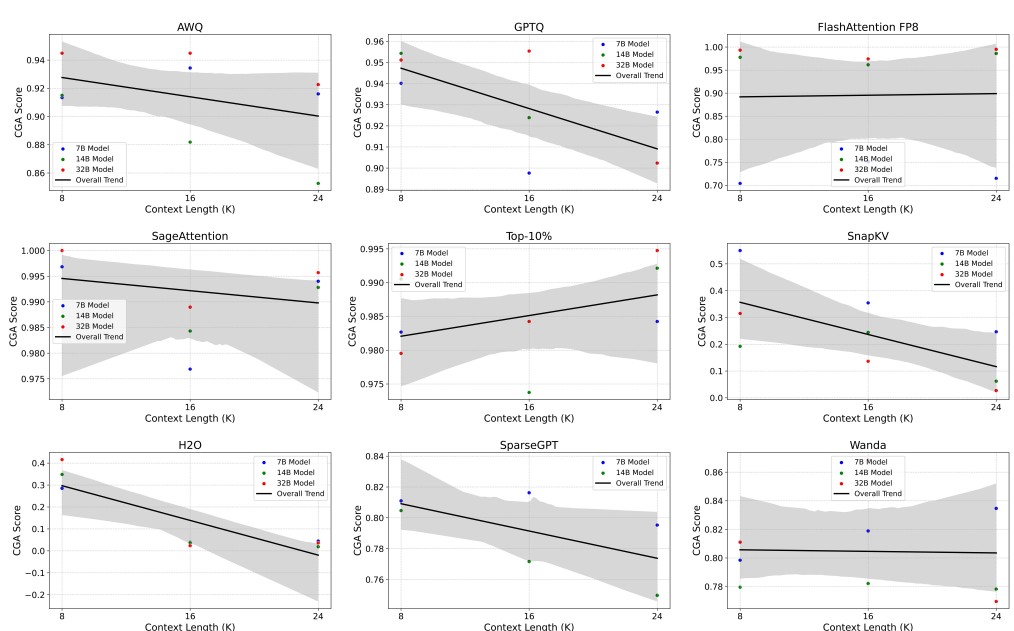

Figure 5: Scaling of compression method fidelity with increasing context length. Methods exhibit distinct behaviors: degradation (quantization (Frantar et al., 2022; Lin et al., 2024), KV cache dropping (Li et al., 2024b; Zhang et al., 2023)), stability (Wanda (Sun et al., 2023b), FlashAttention FP8 (Shah et al., 2024)), or improvement (Top-10% Attention).

manage the computational and memory costs of long-context inference. We evaluated performance under these conditions by stratifying our test data into three tiers based on prompt token count (8K, 16K, and 24K) to assess how the fidelity of each method scales as the context window expands.

As shown in Figure 5, our analysis reveals three distinct performance behaviors as context length increases. Most methods, including all quantization (Frantar et al., 2022; Lin et al., 2024) and KV cache dropping techniques (Zhang et al., 2023; Li et al., 2024b), alongside SparseGPT (Frantar & Alistarh, 2023) and SageAttention (Zhang et al., 2024), exhibits a clear decline in fidelity. In contrast, FlashAttention FP8 (Shah et al., 2024) and the pruning method Wanda (Sun et al., 2023b) maintain stable performance across all context lengths. Uniquely, Top-10% Attention is the only tested method whose fidelity robustly improves with longer contexts. These divergent performance trends underscore that methods within the same category can possess fundamentally different properties, a critical factor for selecting an appropriate compression strategy for long-context applications.

## 5 CONCLUSION AND LIMITATIONS

In this work, we introduced a novel evaluation framework that moves beyond proxy metrics by directly measuring the generative faithfulness of compressed Large Language Models. Our approach, which pairs a new metric, Conditional Generation Accuracy (CGA), with real-world user queries, provides a more reliable assessment of performance degradation. Our comprehensive evaluation of nine mainstream methods established a clear performance hierarchy and revealed distinct scaling laws with respect to model size and context length, offering actionable insights for practitioners. This work is not without limitations. Our strict token-level definition of fidelity may penalize semantically equivalent but tokenically different outputs. Furthermore, our empirical findings are based on a single model family, and the computational cost of our teacher-forcing evaluation is higher than that of standard benchmarks. Future research could explore semantically-aware fidelity metrics and expand the analysis to a broader range of model architectures and compression techniques.

ETHICS STATEMENT

This research adheres to rigorous ethical standards in its methodology and execution. Our work aims to promote transparency and reproducibility within the field of LLM compression by providing a more reliable evaluation framework.

The evaluation dataset was derived from ShareGPT, a publicly available collection of user interactions with language models. To safeguard privacy, we conducted a thorough manual review of the data. During this process, we identified and removed all detectable personally identifiable information. The types of removed information include, but are not limited to, personal identification numbers, API keys, and other documents containing private user details. This sanitization process was critical to ensure the anonymity and privacy of the individuals whose data is represented in the dataset.

All pretrained models used in this study, including the Qwen2.5 family and DeepSeek R1, were accessed and utilized in strict compliance with their respective licensing agreements and terms of use.

To foster continued progress and transparent benchmarking, we have made our evaluation code publicly available. We believe this work represents a positive contribution to the responsible development and assessment of language model technologies.

REPRODUCIBILITY STATEMENT

We are committed to the full reproducibility of our research. All components necessary to replicate our findings are made available to the research community.

The complete source code for our evaluation framework, including the implementation of the Conditional Generation Accuracy (CGA) metric and scripts to run all experiments reported in this paper, is available at https://anonymous.4open.science/r/llm-fidbench. The repository includes detailed instructions for setup and execution.

All experiments were performed on publicly available base models from the Qwen2.5 Instruct series. The compression techniques under evaluation were implemented using their official open source libraries. Our software environment is primarily based on PyTorch, Hugging Face Transformers, and other standard libraries as detailed in the `requirements.txt` file within our code repository. The experiments were conducted on NVIDIA A100 GPUs. We believe these resources provide a clear and complete basis for the verification and extension of our work.

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

## A   USE OF LARGE LANGUAGE MODELS (LLMS)

During the writing of this paper, we utilized LLM solely for language editing to improve clarity and readability. We critically reviewed and revised all AI-generated suggestions to ensure the final text accurately reflects our original intent. All intellectual contributions, including the research design, methodology, analysis, and conclusions, are our exclusive work, and we take full responsibility for the academic integrity of this publication.

## B   CASE STUDY

We provide qualitative examples to illustrate the typical failure modes of several compression methods, revealing performance losses that quantitative metrics often miss.

**AWQ** As shown in Figure 6, models compressed with AWQ can exhibit distracting artifacts such as textual repetition, unhelpful (or "helpless") responses, and generation loops. These issues lead to a perceptible decline in instruction-following ability and overall output quality.

**SageAttention** While generally robust and marketed as nearly lossless, SageAttention can produce entirely meaningless output in specific edge cases (Figure 7). This behavior appears to be a rare artifact rather than a systemic flaw, as the model performs correctly on the vast majority of prompts.

**SnapKV** The performance of SnapKV (Figure 8) starkly contrasts with its published claims. The compressed model consistently fails to generate logically coherent content, rendering it practically unusable for most tasks.

**Wanda** Wanda (Figure 9) severely degrades model performance, producing outputs that are often incoherent. Although its logical consistency is marginally better than SnapKV's, the loss in quality remains substantial.

## C   DATASET PROMPT EXAMPLES

Table 3 provides a selection of representative user prompts from our evaluation dataset, sourced from ShareGPT.

**User Prompt:** I want to help my 10-year old son learn math better. I've started tutoring him recently, but we always end up fighting about it. Are there **ways I can make math learning fun** for his age? Basically teaching him and making him work hard on addition / substraction / multiplication / division / fractions / decimals in a fun way? Give me some ideas please.

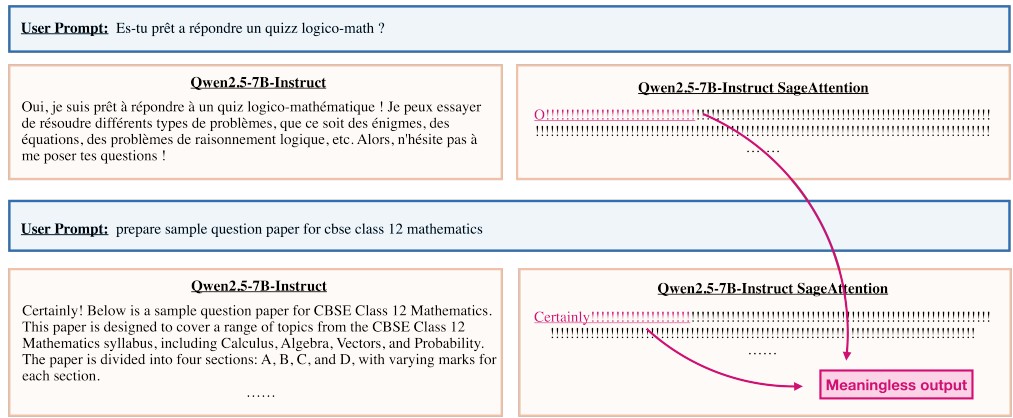

**Qwen2.5-7B-Instruct**

It's great that you're looking for ways to make math learning fun for your son! At 10 years old, he's at a stage where he can grasp more complex concepts, and making the learning process enjoyable can significantly enhance his engagement and understanding. Here are some fun and interactive ideas to help him learn addition, subtraction, multiplication, division, fractions, and decimals:

### 1. **Math Games**
  - **Math Bingo**: Create bingo cards with answers to math problems. Call out problems instead of numbers, and he marks the answers on his card.
  - **Math Jeopardy**: Create a game board with categories like Addition, Subtraction, Multiplication, Division, Fractions, and Decimals. Each category has questions of varying difficulty levels.
  - **Math Card Games**: Use a ... but with a math twist. For example, the ... wins.

  ......

### 9. **Math Challenges**
  - **Daily Challenges**: Set up a daily or weekly challenge where he has to solve a certain number of problems.
  - **Math Competitions**: Participate in online math competitions or join a local math club.

### 10. **Positive Reinforcement**
  - **Praise and Rewards**: Praise his efforts and reward him for completing tasks or solving problems correctly.
  - **Set Goals**: Set small, achievable goals and celebrate when he reaches them.

By incorporating these activities, you can make math learning a more enjoyable and engaging experience for your son. Remember to keep the atmosphere positive and supportive, and be patient as he learns and grows in his math skills.

**Qwen2.5-7B-Instruct AWQ Int4**

Absolutely! Making math fun for a 10-year-old can be a great way to engage them and help them develop a positive attitude towards the subject. Here are some ideas to make learning math more enjoyable:

### 1. **Math Games**
  - **Card Games**: Use a deck of cards to play games like "War" but with a twist. For example, each player draws two cards, adds them up, and the player with the higher sum wins the round.
  - **Board Games**: Games like "Sum Swamp" or "Math Blaster" can be fun and educational. These games often involve basic arithmetic and strategic thinking.
  - **Online Games**: Websites like Khan Academy, IXL, and Coolmath Games offer a variety of games that can help reinforce math skills in a fun way.

### 11. **Math in Technology**
  - **Coding**: Introduce basic coding concepts using platforms like Scratch. Coding often involves logical thinking and problem-solving, which are key math skills.

### 12. **Math in Stories**
  - **Math Stories**: Read books that incorporate math concepts. For example, "The Grapes of Math" by Greg Tang or "The Number Devil" by Hans Magnus Enzensberger.

### 13. **Math in Games**
  - **Math Bingo**: Create bingo cards with math problems. Call out the answers, and the first to complete a row wins.
  - **Math Jeopardy**: Create a Jeopardy-style game with math questions. This can be a fun way to review different topics.
    ......
### 15. **Math in History**
  - **Math in History**: Discuss how math has evolved over time and its importance in history. This can help provide context and make the subject more interesting.
  .......

*2. More generic, often listing broad categories without concrete implementation tips*

*1. Some sections are repetitive*

*3. Outputs endlessly*

Figure 6: Output degradation from AWQ quantization, demonstrating textual repetition and nonsensical content.

**User Prompt:** Es-tu prêt a répondre un quizz logico-math ?

**Qwen2.5-7B-Instruct**

Oui, je suis prêt à répondre à un quiz logico-mathématique ! Je peux essayer de résoudre différents types de problèmes, que ce soit des énigmes, des équations, des problèmes de raisonnement logique, etc. Alors, n'hésite pas à me poser tes questions !

**Qwen2.5-7B-Instruct SageAttention**

O!!!!!!!!!!!!!!!!!!!!!!!!!!!!!!!!!!!!!!!!!!!!!!!!!!!!!!!!!!!!!!!!!!!!!!!!!!!!!!!!!!!
!!!!!!!!!!!!!!!!!!!!!!!!!!!!!!!!!!!!!!!!!!!!!!!!!!!!!!!!!!!!!!!!!!!!!!!!!!!!!!!!!!!!
......

**User Prompt:** prepare sample question paper for cbse class 12 mathematics

**Qwen2.5-7B-Instruct**

Certainly! Below is a sample question paper for CBSE Class 12 Mathematics. This paper is designed to cover a range of topics from the CBSE Class 12 Mathematics syllabus, including Calculus, Algebra, Vectors, and Probability. The paper is divided into four sections: A, B, C, and D, with varying marks for each section.
......

**Qwen2.5-7B-Instruct SageAttention**

Certainly!!!!!!!!!!!!!!!!!!!!!!!!!!!!!!!!!!!!!!!!!!!!!!!!!!!!!!!!!!!!!!!!!!!!!!!!!!!!!!
!!!!!!!!!!!!!!!!!!!!!!!!!!!!!!!!!!!!!!!!!!!!!!!!!!!!!!!!!!!!!!!!!!!!!!!!!!!!!!!!!!!!
......

**Meaningless output**

Figure 7: A rare failure case in SageAttention, where the model produces meaningless output for a specific prompt.

Table 3: Dataset Sketch of User Prompts by Category

| Category | User Prompt |
|---|---|
| Math | what is 1+1? |
| Math | The daily wages for a group of 200 persons have been obtained from a frequency distribution of a continuous variable 'Ă'ŸX'Ă'Ź, after making the substitution U = (X - 130) / 20... Calculate the median, lower and upper quartiles, 3rd decile and 98th percentile. |

Table 3: Dataset Sketch of User Prompts by Category

| Category | User Prompt |
| --- | --- |
| Math | 0/0=0? |
| Math | convex hull problem in 50 words |
| Code | write me a hello world script in c# |
| Code | How do I implement SignalR to broadcast messages in a horizontally scalable .net app? |
| Code | can you write a simple react native app for dream journaling that stores data in json? |
| Code | flutter code to limit items per row |
| Law | Why do people get sentenced for so many years like 1,503 years? |
| Law | What are 10 common events that cause disputes in residential construction? |
| Law | tu es expert en responsabilit'l' civile en droit fran'ğais, aide moi 'ă faire la dissertation portant sur le sujet suivant: 'ń Faut-il supprimer la responsabilit'l' du fait des choses de droit commun ? 'ż |
| Law | You are a juror in the following case: Emily has been charged with assault in the second degree... What is your verdict? |
| Medicine | keto diet side effects |
| Medicine | Topics: Wound management for general practitioners. |
| Medicine | In a study text that teaches medical professionals about the functionality and practical use of the Flotrac device, at least the following 6 points should be chronologically included... |
| Medicine | 單耳常常突然耳鳴是不是要聾了@@ |
| Business | puedes decirme que es un buyer persona '£ |
| Business | write a homepage for translation business |
| Business | Act as a McKinsey technology consultant... you are now responsible of a lecture to a Fortune 500 client who need digitalization. The lecture topic is 'Ă'IJ Maximizing Efficiency with ChatGPT in the age of digitalization: How to Improve Your Workflow and Save Time'Ă'Ï. Please give me an outline for the lecture. |
| Business | Currently our company has 4 core products: OME, OMC, CA and MV... Can you analyse and suggest why are our program head thinking that their program is doing well when the company cashflow is facing issue? |
| French | Quelle est la capitale de la France? |
| French | Comment Jules C'l'sar est-il mort ? |
| French | Donnez trois conseils pour rester en bonne sant'l'. |
| French | D'l'crire la structure d'un atome. |
| Japanese | フランスの首都は何ですか？ |
| Japanese | 原子の構造を説明してください。 |
| Japanese | 健康を維持するための3つのヒントを教えてください。 |
| Japanese | あなたが困難な決断をしなければならなかった時について説明してください。 |
| Chinese | 法国的首都是什么? |
| Chinese | 什么是三原色? |
| Chinese | 渲染一个房屋的三维模型。 |
| Chinese | 给出三个保持健康的提示。 |
| zh2en | Please translate the following paragraph into English. 而我们的银河系如今已经130亿岁了。 |
| zh2en | Please translate the following paragraph into English. 肺癌的体征和症状通常仅在疾病进展时发生。 |

Table 3: Dataset Sketch of User Prompts by Category

| Category | User Prompt |
|---|---|
| zh2en | Please translate the following paragraph into English. 052C/D系列驅逐艦主要是為防空任務設計的，用於保護意義重大的海軍裝備，比如航母和兩棲攻擊艦。 |
| en2zh | 请你把英文翻译成为中文 Our galaxy is 13 billion years old. |
| en2zh | 请你把英文翻译成为中文 Signs and symptoms of lung cancer typically occur only when the disease is advanced. |
| en2zh | 请你把英文翻译成为中文 Let'Ǎ'Źs spend the rest of our lives together, he said. |
| Summarization | Sweden said on Tuesday it was stopping new aid for Cambodia, except in education and research, and would no longer support a reform programme after the main opposition party was outlawed by the Supreme Court at the government s request... (The story adds dropped word after in paragraph 1) Summarize the above paragraph in one sentence. |
| Summarization | U.S. President Donald Trump is strongly committed to working with the European Union toward common objectives of peace and prosperity, U.S. Vice President Mike Pence said on Monday... he continued. Summarize the above paragraph in one sentence. |

## D  RESULTS FOR 14B AND 32B MODELS

This section presents the complete Conditional Generation Accuracy (CGA) results for the larger models, supplementing the 7B model analysis in Section 4.2. Table 4 provides a detailed performance breakdown for Qwen2.5-14B-Instruct, while Table 5 shows the results for Qwen2.5-32B-Instruct. These results provide the underlying data for the scaling analyses in Sections 4.3 and 4.4.

Table 4: Performance comparison of nine compression methods on Qwen2.5-14B-Instruct, measured by CGA. Cell backgrounds are colored on a gradient, with red indicating lower fidelity scores.

| Sub-dataset | AWQ | GPTQ | Flash FP8 | Sage | Top10 | SnapKV | H2O | Sparse | Wanda |
|---|---|---|---|---|---|---|---|---|---|
| fact | 0.88374 | 0.90514 | 0.96535 | 0.98819 | 0.91323 | 0.24536 | 0.57666 | 0.78421 | 0.79285 |
| code | 0.91811 | 0.92362 | 0.94475 | 0.98418 | 0.93071 | 0.45118 | 0.65668 | 0.79055 | 0.80315 |
| en2zh | 0.94420 | 0.96931 | 0.96693 | 0.98740 | 0.96494 | 0.21636 | 0.51924 | 0.82939 | 0.81639 |
| fr | 0.93653 | 0.93885 | 0.96063 | 0.99213 | 0.94887 | 0.32213 | 0.63711 | 0.77904 | 0.79411 |
| law | 0.87905 | 0.91012 | 0.94751 | 0.97638 | 0.90123 | 0.32820 | 0.61104 | 0.73072 | 0.74736 |
| business | 0.88110 | 0.90079 | 0.95793 | 0.98180 | 0.91732 | 0.32362 | 0.63543 | 0.74409 | 0.74488 |
| long16 | 0.88189 | 0.92388 | 0.96149 | 0.98432 | 0.97375 | 0.24409 | 0.03675 | 0.77165 | 0.78215 |
| long24 | 0.85265 | 0.90224 | 0.98616 | 0.99282 | 0.99213 | 0.06220 | 0.01883 | 0.74970 | 0.77824 |
| long8 | 0.91496 | 0.95433 | 0.97778 | 1.00000 | 0.99055 | 0.19213 | 0.34803 | 0.80472 | 0.77953 |
| medicine | 0.91773 | 0.92806 | 0.95984 | 0.98346 | 0.94026 | 0.33656 | 0.65821 | 0.79875 | 0.80909 |
| math | 0.94239 | 0.96762 | 0.95095 | 0.98109 | 0.96898 | 0.22861 | 0.84905 | 0.86317 | 0.88911 |
| sum | 0.84911 | 0.91763 | 0.87194 | 0.98379 | 0.92436 | 0.19489 | 0.71561 | 0.80268 | 0.79995 |
| zh2en | 0.97112 | 0.97157 | 0.98240 | 0.99253 | 0.98973 | 0.23890 | 0.49088 | 0.88655 | 0.87927 |
| jp | 0.91181 | 0.93465 | 0.97008 | 0.98740 | 0.94331 | 0.37743 | 0.55026 | 0.75984 | 0.77323 |
| ch | 0.88851 | 0.91013 | 0.97905 | 0.98593 | 0.92667 | 0.28056 | 0.56939 | 0.73168 | 0.74355 |
| **Knowledge** | 0.890405 | 0.9110275 | 0.9576575 | 0.9824575 | 0.91801 | 0.307935 | 0.620335 | 0.7644425 | 0.773545 |
| **Reason** | 0.93025 | 0.94562 | 0.94785 | 0.982635 | 0.950345 | 0.339895 | 0.65287 | 0.82686 | 0.84613 |
| **Language** | 0.930434 | 0.944502 | 0.971816 | 0.989078 | 0.954704 | 0.287076 | 0.553376 | 0.7977 | 0.80131 |
| **Long** | 0.883166666667 | 0.926816666667 | 0.975143333333 | 0.99238 | 0.984476666667 | 0.16614 | 0.134536666667 | 0.775356666667 | 0.779973333333 |
| **Overall** | 0.908842857143 | 0.931307857143 | 0.965060714286 | 0.986973571429 | 0.950181428571 | 0.274664285714 | 0.496992857143 | 0.787557142857 | 0.795207857143 |

**User Prompt:** The daily wages for a group of 200 persons have been obtained from a frequency distribution of a continuous variable 'X', after making the substitution U = (X - 130) / 20
U =
(X – 130) /
20
-2 -1 0 1 2
No. of
persons
7 50 80 60 3
Calculate the median, lower and upper quartiles, 3rd decile and 98th percentile.

**Qwen2.5-7B-Instruct**

To calculate the median, lower quartile, upper quartile, 3rd decile, and 98th percentile from the given frequency distribution, we first need to understand the distribution of the variable \( U \). The variable \( U \) is a transformation of \( X \) given by \( U = \frac{X - 130}{20} \). This means that the values of \( U \) are centered around 0, and the distribution of \( U \) can be used to find the corresponding values of \( X \).

### Step 1: Determine the cumulative frequencies for \( U \)

The given data is:
- \( U = -2 \): 7 persons
- \( U = -1 \): 50 persons
- \( U = 0 \): 80 persons
- \( U = 1 \): 60 persons
- \( U = 2 \): 3 persons

The total number of persons is 200. We can now calculate the cumulative frequencies:

- For \( U = -2 \): 7
- For \( U = -1 \): 7 + 50 = 57
- For \( U = 0 \): 57 + 80 = 137

.......

**Qwen2.5-7B-Instruct-SnapKV**

To solve for the 50th percentile,1,1, (median), and the 80th percentile, we need to first understand the given data and then proceed step-by-step.

Given data:
- 10th percentile: 10
- 20th percentile: 20
- 30th percentile: 30
- 40th percentile: 40
- 50th percentile (median): 50
- 60th percentile: 60
- 70th percentile:- 80th percentile: 80

To find the 50th percentile (median), 80th percentile, and the 70th percentile, we can use linear interpolation since the percentiles are not given directly.

### 50th Percentile (Median)
The 50th percentile is given directly as 50.

### 80th Percentile
The 80th percentile is given directly as 80.

*incoherent output*

Figure 8: Incoherent and illogical output from SnapKV, highlighting a severe performance collapse.

**User Prompt:** Ask me to answer a high-school math problem. If I say "next", ask me another question. Keep of the questions in a bulleted list. If I type "?", give me the answer.

**Qwen2.5-7B-Instruct**

Sure, let's start with a high-school math problem. Here's your first question:

* What is the value of \( x \) in the equation \( 3x - 7 = 11 \)?

*incoherent and repetitive output*

**Qwen2.5-7B-Instruct-Wanda**

Sure! Let's start with a high school math problem. Here are the questions:

- What is the problem you need help with? Is it algebra, geometry, trigonometry, or something else?

When you're ready, type "next" to proceed to the next question. If you type "?" at any time, I will provide the answer.

Figure 9: Performance loss from Wanda pruning, resulting in largely incoherent and unreliable output.

Table 5: Performance comparison of nine compression methods on Qwen2.5-32B-Instruct, measured by CGA. Cell backgrounds are colored on a gradient, with red indicating lower fidelity scores.

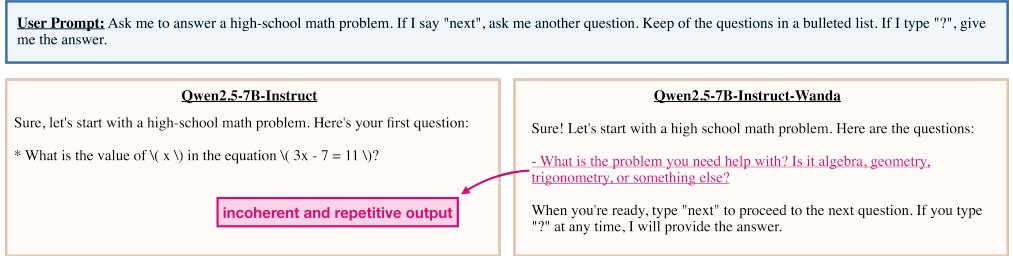

| Sub-dataset | AWQ | GPTQ | Flash FP8 | Sage | Top10 | SnapKV | H2O | Sparse | Wanda |
|---|---|---|---|---|---|---|---|---|---|
| fact | 0.91627 | 0.92525 | 0.98346 | 0.99370 | 0.93526 | 0.08242 | 0.59336 | — | 0.82188 |
| code | 0.94252 | 0.96299 | 0.97586 | 0.99423 | 0.94482 | 0.29665 | 0.69213 | — | 0.85512 |
| en2zh | 0.97161 | 0.95846 | 0.96614 | 0.98268 | 0.98161 | 0.12065 | 0.49183 | — | 0.83436 |
| fr | 0.93682 | 0.94685 | 0.97953 | 0.98898 | 0.94728 | 0.16554 | 0.68004 | — | 0.84629 |
| law | 0.93761 | 0.94058 | 0.96325 | 0.97900 | 0.94567 | 0.21436 | 0.66948 | — | 0.81617 |
| business | 0.92362 | 0.93307 | 0.96870 | 0.98630 | 0.94409 | 0.24724 | 0.67402 | — | 0.79764 |
| long16 | 0.94488 | 0.95538 | 0.97427 | 0.98898 | 0.98425 | 0.13648 | 0.02362 | — | 0.86877 |
| long24 | 0.92263 | 0.98237 | 0.99509 | 0.99570 | 0.99475 | 0.02717 | 0.03616 | — | 0.76954 |
| long8 | 0.94488 | 0.95118 | 0.98333 | 1.00000 | 0.97853 | 0.31496 | 0.41575 | — | 0.81102 |
| medicine | 0.94016 | 0.95669 | 0.96614 | 0.99055 | 0.94016 | 0.20472 | 0.68583 | — | 0.83701 |
| math | 0.97259 | 0.98418 | 0.97638 | 0.98898 | 0.97351 | 0.23589 | 0.83793 | — | 0.84142 |
| sum | 0.90382 | 0.93105 | 0.95956 | 0.99213 | 0.96828 | 0.04705 | 0.72822 | — | 0.84761 |
| zh2en | 0.93741 | 0.95778 | 0.96375 | 0.99375 | 0.98646 | 0.08926 | 0.65552 | — | 0.94595 |
| ch | 0.92441 | 0.93858 | 0.97059 | 0.98894 | 0.93780 | 0.20420 | 0.60144 | — | 0.81654 |
| jp | 0.93663 | 0.93772 | 0.97687 | 0.98904 | 0.94123 | 0.19587 | 0.83286 | — | 0.82983 |
| **Knowledge** | 0.929415 | 0.9388975 | 0.9703875 | 0.9881375 | 0.941295 | 0.187185 | 0.6549225 | — | 0.818175 |
| **Reason** | 0.957555 | 0.973585 | 0.97617 | 0.991605 | 0.961165 | 0.26637 | 0.66503 | — | 0.84827 |
| **Language** | 0.941376 | 0.947878 | 0.973776 | 0.988678 | 0.959476 | 0.155104 | 0.612338 | — | 0.854594 |
| **Long** | 0.937463333333 | 0.963031 | 0.9875633333333 | 0.994893333333 | 0.9861766666667 | 0.1595366666667 | 0.15851 | — | 0.816443333333 |
| **Overall** | 0.939431428571 | 0.946505714286 | 0.97739 | 0.990273571429 | 0.960244285714 | 0.181115 | 0.534783571429 | — | 0.83511 |

