# OpenReview forum: "Beyond Proxy Metrics: A New Evaluation Framework for LLM Compression by Directly Measuring  Generative Faithfulness"
_ICLR.cc/2026/Conference — Submitted to ICLR 2026_

### Official Review · Reviewer_xYyX · 2025-10-31

**Soundness:** 3
**Presentation:** 3
**Contribution:** 1
**Rating:** 0
**Confidence:** 5

**Summary:**

This paper proposes to evaluate compressed LLMs with Conditional Generation Accuracy (CGA) metric, basically comparing average token differences of teacher forced generated tokens over a sequence length. This however was already introduced as the "Divergent Token Metric" in 2024.
They perform analysis on recent datasets and models and compare it to a deepseek judge model.

**Strengths:**

i agree that this way of evaluation should be done more. given that i found *exactly the same metric* in a previous paper, more theoretically and on more use cases explored, the only differing part here is the comparison to a deepseek judge model and a more recent application. (see weaknesses)

**Weaknesses:**

this paper is essentially the "Shared Divergent Token Metric" part of [1], just applied to evaluation only, and on more recent models and datasets.
the authors in [1] demonstrated, even theoretically, that the first ---instead of the share of differing tokens (essentially the "CGA" of the current paper)---, is even more discriminative and better when comparing compressed models. they even applied it during sparsification process to achieve better behaving 'more pruned' models.


on another note, i do not know why one give a statement like 'deepseek score: to approximate human evaluations' - judge model score on its own is fine, statement like this requires a user study.

[1] https://openreview.net/forum?id=AXZLtGlLlBwX

**Questions:**

statement to weakness - did i miss something essential?

---

### Official Review · Reviewer_HUs6 · 2025-10-31

**Soundness:** 3
**Presentation:** 3
**Contribution:** 3
**Rating:** 4
**Confidence:** 4

**Summary:**

This paper introduces Conditional Generation Accuracy (CGA), a metric designed to directly compare the output distributions of the compressed and original models at the token level. Their motivation is stemmed from the hypothesis that proxy metrics such as perplexity (PPL) and standardized benchmarks doesn’t capture all the aspects of the model’s generation capabilities and miss a lot of details to truly evaluate the compressed models, which is very well motivated, and their experiments shows somewhat a promising direction for this new metric.

**Strengths:**

- Clearly written and easy to understand.
- Well motivated. With the size of LLMs increasing and development of various compression methods, it’s a timely study to address how it's best to evaluate the compressed models.
- Experiments covered different model sizes, different context-lengths, which are very practical applications.

**Weaknesses:**

- [**Discussion/Possible Experiment**] How to evaluate a compressed model with this framework in absence of the base/uncompressed model and if there’s no golden response. I believe it’s a very real-world scenario and if not addressed can be seen as one of the major drawbacks for practical evaluation metrics.
    - L397-399: How can one guarantee the determinism of CGA when it involves generation of golden response from uncompressed model?
    - And, if it’s not deterministic, then does the metric needs to be run multiple times and compute average?
- [**Experiment on model choices**] The choice of model to compute “Deepseek Score” might play a role in determining the reliability and alignment with CGA and/or other metrics, thus understanding its influence is important.
- [**Discussion/Possible analysis**] Sec 4.2, 4.3, 4.4 and Key Takeaways: While these patterns are interesting, any hypothesis on why these patterns emerged will further strengthen the paper.
  - At the very least, consider adding practical aspects at the end i.e for applications requiring long context such as summarization, compress using X as our metric suggested that’s the best.
- [**Experiment on different models**] L482-484: While already listed in limitations, consider adding atleast one different model for generalizability purpose.
- [**Discussion/Formal time complexity analysis**] For a question `q`, CGA potentially requires 2 forward passes, one from original model (to build golden response) and the other from compressed model (to actually evaluate the response). If I am not mistaken, this was listed on L482-484 as a limitation. If not, please explain further and consider clarification in paper along with a formal time complexity analysis for computing various metrics such as PPL, CGA etc;

**Questions:**

- L262-263: How do you define challenging prompts? What’s the criteria?
- L264: How difficult it is to evaluate and build benchmark on even longer context lengths? Just wanted to hear your thoughts on encompassing even longer context windows (100K let's say) in your leaderboard benchmark.
- L491-496: Add more details on the cleanup done as it’ll potentially be a leaderboard for models to be evaluated.
- [**Discussion/Clarification**] What are the specific configurations for each of the compressed method? i.e what compression % for Wanda or what’s the group size used in quantized models etc; I would encourage authors to have a section in Appendix with summarizing each compression method along with the configurations used to evaluate.
- [**Discussion/Clarification**] What’s the dataset size? Add more details in Appendix C on the curation and clustering process.

**Possible missing citations**
- L49, L200- The limitations of using PPL and/or standardized benchmark metrics for various compression methods was previously addressed in [1] focusing on evaluating factuality/parametric knowledge aspect of compressed LLM to understand the effects of compression.
- L246-250 is closely related to [2] and have addressed this phenomenon of imitation vs factuality (not specifically targeted for compressed models) along with [1] (specific to compressed models).

1. The Cost of Compression: Investigating the Impact of Compression on Parametric Knowledge in Language Models: https://aclanthology.org/2023.findings-emnlp.349/
2. The False Promise of Imitating Proprietary LLMs: https://arxiv.org/abs/2305.15717

---

### Official Review · Reviewer_ZRvN · 2025-10-31

**Soundness:** 2
**Presentation:** 3
**Contribution:** 2
**Rating:** 2
**Confidence:** 4

**Summary:**

This paper introduces a novel evaluation framework that can be used to compare various model compression methods. The evaluation scheme consists in feeding diverse user prompts into the raw model, and by measuring the next-token prediction accuracy of the compressed model on the generated text.

First, the authors gather real prompts and classify them to ensure their diversity. They proceed to evaluate their method against a model-based pairwise similarity metric that mimics human evaluation. They show that compression methods and families do not behave similarly across evaluations, with some compression schemes clearly outperforming others.

**Strengths:**

This paper tackles the crucial problem of evaluation in model compression approaches. It proposes a novel evaluation framework that complements existing methods. The paper is also well-written and easy to follow.
- **Real-world setup**: I appreciate the fact that the evaluation method is based on real prompts. It can thus be extended and improved over time by adding more user prompts, i.e. low-supervised data, which is not easily possible for e.g. QA benchmark where high-quality questions need to be curated. It also allows the authors to assess model degradation on a dataset with controllable diversity, and pinpoint domains where it is more important.

**Weaknesses:**

I have concerns about the methodology of the paper and about how it compares rather different compression methods. I also find that some crucial experimental points are not properly reported by this paper, which hurts both reproducibility and soundness.
- **Conceptual limitations**: As mentioned in the conclusion, the presented method is rather lexically (or tokenically, in the authors' words) grounded, as the CGA scores measures the ability of a compressed model to mimic the behavior of its raw counterpart. This raises several limitations. First, two answers may be very similar lexically but semantically extremely different, e.g. a reasoning trace that is very lexically similar but concludes with the wrong answer, which would be poorly captured by the CGA metric. Second, it is not clear that the goal of model compression is to minimize the gap in generated outputs before and after compression. A prompt can be answered in different ways that are all satisfactory, and a compressed model might still be relevant even when its answer differs from the raw model. Some compression methods claim performance improvements over the raw baselines. In that context, comparing CGA with the mentioned BERTScore-based baseline would provide useful insights.
- **Lack of details on human-like evaluation**: It can be argued that model-based evaluation is not a perfect proxy for human assessment, but the human cost of such assessment being prohibitive in most cases, this choice can be justified. However, contrarily to what is usually done in human evaluation, the authors provide very few details about their methodology: what prompt was used? Would another judge model agree with DeepSeek? Were generation parameters tuned properly? This is particularly important, as the used prompt or DeepSeek could also be specifically looking at lexical similarity, in which case the correlation comparison between CGA and other methods is unfair.
- **Comparison of evaluation methods**: Comparing such different compression method is not very relevant. Every method family has a different purpose and a different efficiency impact. In that context, the latency and memory improvements yielded by each method in different contexts is a crucial comparison point to better illustrate their efficiency. In other words, the question that should be asked is: for a given target acceleration/memory gain, what performance can each method retain? This discussion is missing in the paper, and raises questions about the relevance of the results. As a result, the conclusion about a "hierarchy of methods" lacks meaning: quantization can reduce memory usage and latency, but it will still saturate VRAM for long sequences, while KV cache compression can reduce long-context memory usage but it is likely to perform poorly on short prompts. Moreover, it is unclear how the Top-10% Attention method works, as it is never clearly explained in the paper; my guess is that only the top-10% attention weights are considered at every step, which would imply that all KV items are kept in-memory across generation, and that query-key products are computed for all pairs. If so, it is not clear what practical gain this approach yields, as both memory usage and FLOPS are roughly equivalent to the raw model behavior.

**Questions:**

- How does the top-10% attention method work?
- What prompt did you use for DeepSeek scoring?
- What parameters did you use for every compression method (especially for KV cache compression)? How did each method perform in terms of latency and memory? How does each method compare from the efficiency vs. performance trade-off viewpoint?

---

### Official Review · Reviewer_mdHr · 2025-11-11

**Soundness:** 3
**Presentation:** 3
**Contribution:** 2
**Rating:** 6
**Confidence:** 3

**Summary:**

The paper identifies that existing evaluation strategies for compressed LLMs cannot reflect performance in real scenarios. To address this, the paper proposes CGA, an evaluation framework that directly measures the ability of a compressed model to replicate the original model's prediction. The evaluation results on various model sizes and context lengths reveal that the quantised model's performance may not increase as the model size increases.

**Strengths:**

The paper identifies an important issue in open-ended response generation evaluation. The proposed CGA uses the original model's output as a reference to measure the compressed counterpart, which is simple and effective, without relying on heuristic metrics such as edit distance and model-based metrics like BERTScore.

The evaluation is comprehensive, including various model sizes, context lengths, and compression methods, e.g., lower precision, quantisation, pruning, kv cache dropping, and sparse attention.

The conclusion regarding the performance of different compression methods is interesting and may provide some insight for future research.

**Weaknesses:**

Though the method is proposed to address the unreliable evaluation of open-ended questions in real applications, the teacher-forcing nature may not truly reflect the real performance, such as long-form generation. Will the compressed model output repeated tokens even though in a teacher-forcing setting?

The issue of teacher-forcing may also introduce bias in the evaluation. For example, in the top-10% sparse attention model will always condition on a golden context to select top-10% kv, which is near golden due to the inherent sparsity property of Transformers. It may exaggerate its performance.

**Questions:**

see in weakness

---

### Meta-Review · Area_Chair_iXNZ · 2026-01-03

**Summary:**

The paper provides a well-motivated and timely framework (CGA) for evaluating model compression in open-ended generation by using the original model's output as a reference, supported by comprehensive experiments across various model sizes, context lengths, and compression techniques.

There are multiple important concerns raised by reviewers:
1. Novelty and Methodology: Reviewers noted that the "CGA" metric appears identical to the "Shared Divergent Token Metric" from existing literature. Furthermore, the use of teacher-forcing may exaggerate performance and fail to reflect real-world issues like token repetition in long-form generation.

2. Conceptual Alignment: Since the metric is strictly lexical (token-based), it may penalize compressed models that provide semantically correct but lexically different answers, or fail to catch cases where a single token change completely alters the meaning or reasoning of a response.

3. Evaluation Reliability: The "DeepSeek Score" used as a proxy for human judgment lacks sufficient detail (e.g., prompts used, inter-judge agreement) and a user study to prove it truly aligns with human preferences.

There are other concerns like evaluation fails to account for the efficiency-performance trade-off and the proposed approach is computationally expensive and makes the framework difficult to use in real-world scenarios

**Reviewer Concerns:**

The authors didn't respond to the concerns from the reviewers.

**Reviewer Scores:**

n/a

---

### Decision · Program_Chairs · 2026-01-26

Reject